# Do Primary School Children Benefit from Drop-Jump Training with Different Schedules of Augmented Feedback about the Jump Height?

**DOI:** 10.3390/sports10090133

**Published:** 2022-09-02

**Authors:** Christian Leukel, Sabine Karoß, Florian Gräßlin, Jürgen Nicolaus, Albert Gollhofer

**Affiliations:** 1Department of Sport Science, University of Freiburg, 79117 Freiburg, Germany; 2Bernstein Center Freiburg, University of Freiburg, 79104 Freiburg, Germany; 3Department of Everyday Culture, Sports and Health, University of Education, 79117 Freiburg, Germany; 4Primary School Kirchzarten, 79199 Kirchzarten, Germany

**Keywords:** exercise, functional performance, physical performance, training, motor behavior

## Abstract

In children, the training of jumps leads to improved jumping and running performance. Augmented feedback about the jump height is known to facilitate performance improvements in adults. In the present study, the impact of augmented feedback on jumping performance was investigated in 4th grade primary school children executing drop-jump training for 8 weeks (24 sessions, 3 times/week). Ten children (eight males, two females, aged 9.6 ± 0.3 years), received feedback for 8 weeks, and 11 children (nine males, two females, aged 9.5 ± 0.2 years) received feedback only during the last 4 weeks. Drop-jumps training was integrated in physical education classes. Drop-jump and countermovement-jump heights were improved after 24 training sessions (*p* < 0.01 for both types of jumps in both groups). Ground contact times of drop-jumps were quite long (>200 ms) and not altered by training, and the reactive strength index of drop-jumps was between 0.75 and 1.5 in most children. Augmented feedback did not facilitate jumping performance like in previous studies with adult participants. In contrast, withholding augmented feedback during the first 4 weeks of training was associated with a reduction in jumping performance (*p* < 0.01 for drop-jumps, *p* < 0.05 for countermovement-jumps). Finally, improvements did not transfer to functional motor tasks containing jumps. According to the costs and outcomes we do not recommend drop-jump training with augmented feedback about the jump height for 4th grade physical education classes.

## 1. Introduction

Feedback is a crucial determinant for performance improvements induced by learning. Behavior is updated by the processing of inputs from the sensors of the body [1,2]. With regard to the source of the information, feedback can be subdivided into two entities, namely extrinsic and intrinsic feedback [3]. Extrinsic or augmented feedback describes information that originates from a source outside of the learner. In a school setting, this source is usually the teacher who recognizes movement errors and reports these errors back to the student. Intrinsic feedback, on the other hand, describes information that originates from the body of the learner [4]. Augmented feedback can lead to substantial performance improvements that surpass improvements reached with intrinsic feedback [3,5].

Augmented feedback can facilitate performance improvements in motor learning, but its effect depends on the skill [4]. For jumping, a facilitatory effect from augmented feedback on performance has been repeatedly [6,7,8,9]. Jumping is relevant in sports, e.g., athletics, gymnastics, and ball games [10,11]. In a study by Keller and colleagues [7], applying augmented feedback about the achieved jump height during four weeks of drop-jump training led to significant increases in drop-jump height. In a very recent study, Leukel and Gollhofer [8] extended these previous findings and applied augmented feedback in a practical setting. Subjects performed catch and shoot exercises with a basketball and received feedback about the jump height after they jumped and shot the ball at the basket. In this study, it was demonstrated that augmented feedback can improve jumping performance even when the focus is not on jumping but on basketball training [8]. Taken together, previous investigations clearly demonstrate facilitated jumping performance with augmented [7,8,9].

Thus far, adults have been targeted in studies about augmented feedback and jumping. To the best of our knowledge, in children the impact of augmented feedback on jumping performance is yet unknown. Jumping is considered a fundamental motor skill in children [12,13,14]. Prepuberal children displayed increased jumping and running skills after plyometric training [15], and increased drop-jump heights with a higher training status in sports like gymnastics and athletics [16,17]. In the present study, we aimed at investigating possible facilitatory effects from augmented feedback about the jump height compared to no feedback in prepuberal children who undergo drop-jump training. Therefore, a previously developed experimental protocol consisting of drop-jump training was utilized, which in adult participants proved to be effective in facilitating jumping performance [7]. In contrast to previous studies, the current study was conducted in a school setting, in physical education classes, and not in the laboratory. The reason for this change of the setting concerned the importance of reliable feedback for performance improvements in school [18,19]. Facilitating children’s jumping achievements through reliable feedback can be of practical interest in the physical education classroom. We tested 4th graders (highest grade in elementary school in the State of Baden-Württemberg in Germany) and not younger children because compliance can be expected at this age, meaning that the children are cognitively able to carefully execute the training and test sessions.

The children in the current study performed drop-jump training over a period of 8 weeks. According to previous findings about positive associations between drop-jump performance in prepuberal children and expertise levels in sports where jumps are regularly executed (e.g., gymnastics, athletics) [16,17], we hypothesized that the children in the present study would show increased jump heights of drop-jumps and countermovement-jumps after training. Further, according to the facilitatory effect from augmented feedback on drop-jump performance in adults [7,8], and findings in children that performance can be facilitated through augmented feedback about task results (i.e., knowledge of result) [20,21,22], we hypothesized that augmented feedback would facilitate increases in jump height in the present study. In addition to the testing of drop-jumps and countermovement-jumps, changes in performance of functional motor tasks were assessed. Based on known transfer effects of improved jumping performance to other motor skills [15], we hypothesized that changes in jumping behavior in the present study would transfer to motor tasks containing jumps. Positive transfer from jumping to motor tasks was analyzed with linear regression models and also changes in performance of the motor tasks.

## 2. Materials and Methods

### 2.1. Participants

Twenty-one healthy children participated in this study (Table 1). The children were recruited from a local primary school we had been collaborating with before, and were all 4th graders in the same class. Recruiting a whole class, in contrast to selecting individuals from different classes, was the only way for the school to meet the demands of the training and testing times of the experimental protocol. Recruiting the whole class was also the only way for us to conduct this study, because of the administration and supervision required to conduct the training and testing. Exclusion criteria were neurological disorders and/or injuries/diseases of the musculoskeletal system, which were not met by any of the children. Thus, all children of the class were allowed to participate. Males and females were allocated to the intervention group INT (8 males, 2 females, aged 9.6 ± 0.3 years) and the waiting-control group CON (9 males, 2 females, aged 9.5 ± 0.2 years). Written informed consent was obtained from all parents (or legal guardians) and also the children. The study was conducted according to the guidelines of the Declaration of Helsinki (latest revision in Fortaleza) and approved by the local ethics committee (21-1280).

### 2.2. Experimental Design

The children participated in physical education classes three times a week, on Mondays, Tuesdays, and Thursdays. These were the times when the children had physical education lessons. Each lesson lasted for 45 min, which is the duration of a physical education class in the German school system. Drop-jump training was integrated in the physical education classes. There were 24 sessions over a period of 9 weeks (1 week of holidays between the first and second block of 4 weeks). From sessions 1 to 12, half of the children (the intervention (INT) group) received augmented feedback about the jump height. From sessions 13 to 24, all children, including those in a waiting control group (the CON group) received augmented feedback about their jump height (Figure 1). Children in the CON group received feedback during the second half of training because this was demanded by the ethics committee approving this study. The alternative, receiving no feedback at all, was not permitted. Performance changes of drop-jumps and countermovement-jumps were assessed at three points in time. The pre-test took place 3 days before the first drop-jump training. The mid-test took place 3 days before the 13th training session, and the post-test took place 2 days after the final training session. These tests were conducted in the training science laboratory located at the University of Education in Freiburg. For the pre-, mid-, and post-test, all children were tested on a single day, and therefore visited the laboratory as a group. Single children were then called to the laboratory and tested while the others were supervised by students and played in the gym at the University campus. Note that the children were not allowed to exercise until 5 min before they were called to the laboratory. They were allowed to play boardgames, read books, do homework, and/or watch a movie. Five minutes before moving to the lab, the child had to run for 3 min in the gym and had to perform low intensity two-legged hopping for another approximately 1 min. This was supervised and controlled for by an experimenter. The gym was located in the same building as the lab. A single measurement (one child) lasted for approximately 8 min. Thus, it took approximately 3 h to complete all tests.

In addition to the tests in the laboratory, performance of functional motor tasks was assessed at the local primary school. The functional motor tasks consisted of rhythmic jumping, standing long-jump, and scissors-jump. We chose these tasks for the following reasons: rhythmic jumping is similar to drop-jumps from low heights and adds the aspects of repetition and rhythmic coordination [23,24]. Standing long-jump is a functional motor task with similar features than a countermovement-jump, which was also tested in the present study. According to previous studies [8,25], it was assumed that performance levels in individuals will vary similarly for drop-jumps and countermovement-jumps. The scissors-jump was included because the physical education teacher of the tested class recommended this task as it was known by the children and part of previous lesson plans. According to the timing of tests in the laboratory, performance changes of functional motor tasks were assessed 2 days before the first training session, 2 days before the 13th training session, and 3 days after the final training session, respectively (Figure 1).

### 2.3. Procedures

#### 2.3.1. Performance of Drop-Jumps and Countermovement-Jumps

The order of the children entering the lab was the same in the pre-, mid-, and post-test. In the pre-test, a countermovement-jump and a drop-jump were demonstrated outside of the lab by an experimenter prior to the measurement, so that the children got familiar with the jumping procedure. The children were then allowed to practice the jumps while being supervised and corrected by the experimenter. All children were supervised by the same experimenter.

In the lab, ground reaction forces (AccuGait^®^, AMTI, Watertown, USA, sampling rate of 1 kHz) were recorded for 10 consecutive countermovement-jumps and 10 consecutive drop-jumps. The falling height was 30 cm, analogous to our previously published study [7] from which the experimental design was borrowed. Only low falling heights are recommended in prepuberal children [26]. Each child executed the two types of jumps in the pre-, mid-, and post-test in the same order, i.e., started either with drop-jumps or countermovement-jumps. The order of the jumps was balanced between groups. Before recording blocks of 10 countermovement-jumps/drop-jumps, the children performed 5 corresponding submaximal warm-up jumps (5 countermovement-jumps or 5 drop-jumps, respectively). The children were asked to jump as high as possible, and perform short contact times while drop-jumping, but no further advice was given with regard to knee, ankle or hip angles and duration of ground contact times during jumping. The children kept their hands at the right and left ilium. Jumps had to be performed barefooted. The children had to rest for at least 5 s between successive jumps and 2 min after the recording of 10 jumps was completed. Augmented feedback was not provided in the pre-, mid-, and post-test.

#### 2.3.2. Functional Motor Tasks

Performance of the three tasks (rhythmic jumping, standing long-jump, and scissors-jump) was tested in the primary school gym. Each of the tasks was supervised by an experimenter, and the assignment of the experimenters to the tasks was the same for the pre-, mid-, and post-test. Three groups of children moved together between task stations, and all of the children in a group completed the task before they moved to the next station. The children wore the same shoes in the pre-, mid-, and post-test.

For rhythmic jumping, the children had to jump over 10 hurdles, which were placed parallel to each other and with equal distances of 50 cm on the floor. The height of one hurdle was 15 cm. The children were instructed to complete the task as quickly as possible. In the pre-, mid-, and post-test, two practice runs were performed prior to two test runs. Performance referred to the elapsed time in seconds between the start of the movement until the crossing of the 10th hurdle was completed. The elapsed time was measured with a handheld stopwatch. A run had to be repeated if a child knocked down a hurdle. This happened three times in this study.

For the standing long-jump, the distance in centimeter between the starting line and the back of the heel was measured with measuring tape. The trial was skipped in case a child fell backwards after landing and touched the floor with the hand(s) for support. The children were instructed to jump-off and land with both feet, but no further instruction was provided concerning the jumping technique. The children were allowed to practice three times before two test trials were recorded.

For the scissors-jump, the children had to jump over an elastic rope that was fixed between two poles placed at two opposing corners of a floormat. All children started at a height of 50 cm, which was increased in steps of 10 cm if the elastic rope was not touched while jumping over it. If the rope was touched, the children were allowed to repeat the jump over the same height, and after a total of two unsuccessful attempts, they had to stop jumping. Performance referred to the maximum height that was successfully (i.e., without touching the rope) crossed. The children had practiced the scissors technique with their physical education teacher prior to the pre-test.

#### 2.3.3. Training

Each physical education class lasted for 45 min. Figure 1 depicts the total number of completed sessions for each participant. All children had to perform 10 drop-jumps from a 30 cm platform in each session, with pauses of at least 10 s between trials, as in our recent study [8]. The children kept their hands at the right and left ilium. The children were performing drop-jumps in small groups of 3 to 7 children in a consecutive order, and thus the time for one child to complete the jumps in each session took around 5 to 10 min. They were asked to jump as high as possible and with short ground contact times. Flight times were measured with a light barrier (Sick, Waldkirch, Germany) with the receiver and transmitter being placed 1.5 m apart. The jump height was calculated using a LabView-based algorithm accordingly to the formula: 1/8 × g × t^2^ (g refers to the acceleration of gravity and t refers to the duration of the flight phase). The result (jump height in centimeter) was presented approximately 1 s after completion of the jump and was visible on a computer screen in front of the children. The display time of the feedback was 4 s. The drop-jumps were integrated in the physical education classes, which means that children performed the jumps at times during the physical education class that fitted to the lesson plan. Three experimenters supervised the jumps in all sessions, and procured that all children completed the required 10 jumps in each session using tally sheets.

### 2.4. Data Analysis and Statistics

Individual mean values from the 10 jumps were calculated for jump heights and ground contact times. The reactive strength index for drop-jumps, indicating reactive strength capacity, was calculated according to Sialis [27]: individual mean values of the time airborne were divided by the individual mean values of ground contact times in seconds. For rhythmic jumping, the lowest value was selected from the two trials and used for further analysis. For standing long-jump, the largest value was selected from the two trials. For scissors-jump, the maximum height was chosen. The corresponding values were analyzed using repeated-measures ANOVAs with the factors GROUP (INT and CON group) and TIME (pre-, mid-, and post-test). Post hoc Student’s *t*-tests were carried out if factors or combination of factors reached significance. Partial eta-squared values (η^2^) were calculated according to Bakeman [28] to estimate the effect sizes of significant results of the ANOVAs. Greenhouse–Geisser corrected values were calculated and are reported in case sphericity was violated based on Mauchly’s test of sphericity.

Multiple linear regression analyses were calculated to test if performance of drop-jumps and countermovement-jumps could predict children’s’ performance of the three tasks, namely rhythmic jumping, standing long-jump, and scissors-jump, respectively. All values were z-transformed before calculating the models.

Normal distribution and homogeneity were confirmed for all data sets by the Kolmogorov–Smirnov test and Levene test, respectively. The level of significance was set to *p* < 0.05 for all test. Offline data analyses and statistical analyses were performed using R and RStudio software (RStudio Inc., Boston, MA, USA).

## 3. Results

### 3.1. Jump Heights

Jump heights of countermovement-jumps and drop-jumps are depicted in Figure 2.

For drop-jumps, the repeated measures ANOVA yielded a significant effect for TIME (*F*_2,38_ = 57.5, *p* < 0.01, η^2^ = 0.26) and for GROUP × TIME (*F*_2,38_ = 11.0, *p* < 0.01, η^2^ = 0.06). The factor GROUP did not reach significance (*F*_1,19_ = 0.32, *p* = 0.58). Post hoc *t*-tests were calculated because of significant GROUP × TIME interactions. They yielded a significant decrease in jump height from pre to mid in the CON group (*p* < 0.01; t = −3.8; 95% confidence interval (CI = −4.8, −1.2). The jump height did not change from pre to mid in the INT group (*p* < 0.65; t = −0.5; 95% CI = −1.6, 1.0). Jump height increased from mid to post in the INT group (*p* < 0.01; t = −4.1; 95% CI = −4.1, −1.2) and the CON group (*p* < 0.001; t = −14.8; 95% CI = −8.0, −5.9). Jump height also increased from pre to post in the INT group (*p* < 0.01; t = 4.1; 95% CI = 1.1, 3.7) and the CON group (*p* < 0.001; t = 5.1; 95% CI = 2.2, 5.7).

For countermovement-jumps, the repeated measures ANOVA yielded a significant effect for TIME (*F*_2,38_ = 106.2, *p* < 0.01, η^2^ = 0.24) and for GROUP × TIME (*F*_2,38_ = 10.6, *p* < 0.01, η^2^ = 0.03). The factor GROUP did not reach significance (*F*_1,19_ = 0.19, *p* = 0.67). Post hoc *t*-tests were calculated because of significant GROUP × TIME interactions. They yielded a significant decrease in jump height from pre to mid in the CON group (*p* < 0.01; t = −3.3; 95% CI = −2.3, −0.5). The jump height did not change from pre to mid in the INT group (*p* < 0.64; t = −0.48; 95% CI = −0.8, 0.5). Jump height increased from mid to post in the INT group (*p* < 0.001; t = −5.6; 95% CI = −3.3, −1.4) and the CON group (*p* < 0.001, t = −19.8, 95% CI = −5.3, −4.2). Jump height also increased from pre to post in the INT group (*p* < 0.001, t = 6.4, 95% CI = 1.4, 3) and the CON group (*p* < 0.001, t = 7.9, 95% CI = 2.4, 4.3).

For drop-jump and countermovement-jump, the statistical results yielded higher jump heights after 24 training sessions, but no facilitatory effect from augmented feedback on jump height. A possible reason concerning the latter could be that the sample size was too low. To test this possibility, bootstrapping (1000 sweeps) of the INT group sample was performed, and mean values of the pre- and mid-test were compared. Differences of the mean values (pre minus mid) of the sweeps were compared to the differences of the actual mean values (pre minus mid), in that the number of times were counted in which the result of the sweeps was larger than the result of the actual data [29,30]. The resultant *p*-value from this analysis was 0.87 for drop-jump height, and 0.93 for countermovement-jump height, indicating that a larger sample size would most likely not have overturned the results.

### 3.2. Ground Contact Times

Ground contact times of drop-jumps are depicted in Figure 2.

The repeated measures ANOVA yielded no significant effects for GROUP (*F*_1,19_ = 1.12, *p* = 0.3), TIME (*F*_2,38_ = 3.0, *p* = 0.06) and for GROUP × TIME (*F*_2,38_ = 0.01, *p* = 0.98).

### 3.3. Task Performance

Performances of the three tasks are depicted in Figure 3. Please note that two subjects had to be excluded from the analyses, one in the INT group (subject ID: 9) and one in the CON group (subject ID: 9). The subject in the INT group was unable to attend the mid-test, the subject in the CON group was unable to attend the post-test.

For rhythmic jumping, the repeated measures ANOVA yielded a significant effect for TIME (*F*_2,34_ = 19.3, *p* < 0.01, η^2^ = 0.20). Neither the factor GROUP (*F*_1,17_ = 0.00, *p* = 0.98) nor the factor GROUP × TIME (*F*_2,34_ = 11.0, *p* = 0.94) reached significance. Post hoc *t*-tests were calculated because of significant effect for TIME. They yielded no significant change of performance between pre to mid for the INT group (*p* = 0.90, t = 0.12, 95% CI = −20.5, 22.9) and the CON group (*p* = 0.74, t = 0.3, 95% CI = 16, 11.9). From mid to post, performance did significantly increase in the INT group (*p* < 0.05, t = 3.3; 95% CI = 10.7, 60.1) and the CON group (*p* < 0.001, t = 5.3, 95% CI = 12.3, 30.8). From pre to post, performance also did significantly increase in the INT group (*p* < 0.001, t = −5.6, 95% CI = −48.1, −20.2) and the CON group (*p* < 0.01, t = −3.3, 95% CI = −40, −7.2).

For standing long-jump, the repeated measures ANOVA yielded no significant effect for GROUP (*F*_1,17_ = 0.7, *p* = 0.41), TIME (*F*_2,34_ = 2.6, *p* = 0.09), and GROUP × TIME (*F*_2,34_ = 1, *p* = 0.38).

For scissors-jump, the repeated measures ANOVA yielded no significant effect for GROUP (*F*_1,17_ = 0.02, *p* = 0.88), TIME (*F*_2,34_ = 3.1, *p* = 0.06), and GROUP × TIME (*F*_2,34_ = 0.01, *p* = 0.99).

For rhythmic jumping, the results of the linear regression analysis yielded that the model explained 39% of the variance and significantly predicted rhythmic jumping performance (*F*_2,54_ = 17.14, *p* < 0.01). While performance of countermovement-jumps contributed significantly to the model (β = −0.46, *p* < 0.05), performance of drop-jumps did not (β = −0.18, *p* = 0.39).

For standing long-jump, the results of the linear regression analysis yielded that the model explained 23% of the variance and significantly predicted standing long-jump performance (*F*_2,54_ = 8.2, *p* < 0.01). While the performance of countermovement-jumps contributed significantly to the model (β = 0.55, *p* < 0.05), the performance of drop-jumps did not (β = −0.08, *p* = 0.72).

For scissors-jump, the results of the linear regression analysis yielded that the model explained 12% of the variance (*F*_2,54_ = 3.9, *p* < 0.05). None of the jumps contributed significantly to the model (countermovement-jumps: β = 0.28, *p* = 0.26; drop-jumps: β = −0.08, *p* = 0.76).

## 4. Discussion

Jumping performance and ground contact times.

Jumping plays an important role in sports. The positive impact of repeated jumping, mainly in the form of plyometric training, on jumping skills has been documented for many target groups, including children [15,31]. In line with these previous findings, jump heights of both drop-jumps and countermovement-jumps were significantly increased after 24 training sessions in the present study. A novel aspect of the current study is that drop-jump training was implemented in primary school, integrated into regular physical education classes. Development of sensorimotor skills is one of three process-oriented goals in school curricula in the State of Baden-Württemberg in Germany. The results about improvements in jumping performance are thus in accordance with these objectives.

In contrast to jump heights, ground contact times did not change by training in the present study. In general they were quite long (>200 ms), considering typical values of below 200 ms for stretch-shortening cycle contractions [10]. The reactive strength index (i.e., dividing physical work by contact times,) was similar to results of a recent study in which children also performed drop-jumps from 30 cm falling height [25]. In the current study, it was in between 0.75 and 1.5 in most subjects. The reactive strength index was reported to be around 2.5 in a large sample of adult participants [27]. This discrepancy indicates that the efficiency of the stretch-shortening cycles was lower compared to adults, most likely because of reduced stiffness properties of the tendomuscular unit in prepuberal children [32,33].

### 4.1. Augmented Feedback

We hypothesized that augmented feedback about the jump height would facilitate performance improvements as it was documented in several previous investigations in adult participants performing training over three to four weeks [7,8,9]. However, the jump height of drop-jumps and countermovement-jumps did not increase between the pre-test and the mid-test in the INT group. A possible explanation for the difference between the previous and current findings is that children could not use the rather abstract information about the jump height properly for adapting movement parameters that allow for higher jumps. As a consequence, children may depend on information that directly links movement parameters with preferred outcome (e.g., “if you place your foot more to the right, you will jump higher”) more than adults do.

When feedback was provided to both groups, INT and CON, in the second phase of training, jump height was significantly enhanced in the post-test. Because both groups received feedback in this phase, it is impossible to infer its impact on jumping performance. Augmented feedback and/or repeated exercise, the latter referring to the practice times from sessions 13 to 24, can be the cause of the increased jump height.

Withholding children in the CON group from receiving augmented feedback in the first phase of training led to a significant reduction in jump height. A possible explanation causing the deteriorated performance is deprived motivation in these children. Indeed, motivation is considered a crucial performance-enhancing component of augmented feedback [4,9,21]. Both groups, INT and CON, trained in parallel in the physical education classes in the present study, which means that, for subjects in the CON group, it was possible to observe that feedback was provided to their peers. Some children in the CON group actually complained about the fact that only selected members of the class (i.e., the INT group) were allowed to receive feedback. This may have led to disappointment and possibly frustration, which, as a consequence, caused submaximal jumping performance even though the children were encouraged to maximize their jump height in training and test sessions. A practical consequence is not to prevent children from receiving augmented feedback when noticing that feedback is provided to their peers.

### 4.2. Transfer to Motor Tasks

We expected positive transfer of increased jump heights to functional motor tasks, namely rhythmic jumping, standing long-jump, and scissors-jump. In fact, only the performance of rhythmic jumping was significantly increased in both the INT and the CON group in the post-test. This is in contrast to results from a previous study in which jumping distance (standing long-jump) was significantly enhanced after plyometric training [34]. The contrasting results might be explained by a different number of jumps per session and/or the different training regime. Participants in the study of Almeida et al. [34] executed between 50 and 120 jumps per session, compared to only 10 jumps in our study. Further, different types of jumps, including lateral jumps and squat jumps, were exercised in the study of Almeida et al. [34], not solely drop-jumps as in the present study. Our results indeed support the notion that the training of drop-jumps is not significantly influencing performance of standing long-jumps. Countermovement-jumps but not drop-jumps drove the model from linear regression for standing long-jumps. This suggests that the training of drop-jumps does not transfer to standing long-jump.

Results from the multiple linear regression analyses in the present study also raise concerns about the relevance of drop-jump training for improvements in rhythmic jumping. As with standing long-jumps, for rhythmic jumping only countermovement-jumps but not drop-jumps significantly contributed to the model. Therefore, improvements in rhythmic jumping may not have been driven by drop-jump training at all, but rather be caused by repeated exposure to the task. According to the null finding for standing long-jump and scissors-jump, and the yet unclear reason underlying the performance improvement in rhythmic jumping, positive transfer effects from drop-jump training to functional motor tasks are concluded to be marginal at best. This means that performance improvements of the tested motor task cannot be expected from the type of drop-jump intervention applied in this study.

## 5. Limitations

There are several limitations. First, it is still unclear if a longer exposition to augmented feedback for 8 weeks would facilitate performance improvements of jumps. The CON group in our study also received augmented feedback during the second half of training due to demands from the local ethics committee approving this study. Second, subjects in previous studies typically performed a larger number of jumps in a single session compared to the children in the present study, and this makes it difficult to compare training outcomes. In our study, we could not increase the number of jumps in a single session because training was integrated in physical education classes. The jumping exercises were just added to the regular program and not the main focus. Third, we cannot exclude that the outcomes after the second phase of training contained delayed effects from the first training period. This could be solved by introducing a longer break in between the two phases of training than just one week which was due to holidays in the present study. Forth, we tested a special kind of jump training with solely drop-jumps because of previous experiences with these types of jumps and the setup. The results have to be viewed according to this situation. Other forms of jump training may lead to different results. Fifth, we only tested 4th graders in primary school. The results thus do not apply to primary school children in general. It is possible that younger children would react differently to training [35]. Sixth, transfer effects are constrained to the tasks applied in this study. It is certainly possible that performance of other motor tasks could be affected differently by the drop-jump training. Seventh, knowledge of results was provided to the children in the present study. Effects resulting from knowledge of performance were not tested. Knowledge of performance in this particular study could relate to changes of kinematic variables (e.g., knee and hip flexion/extension) while performing the jumps, with the goal to improve the jumping technique. This information may have a different effect on jumping performance than the knowledge of result about the jump height.

## 6. Conclusions

This study revealed increased drop-jump and countermovement-jump height in 4th grade primary school children after 24 sessions of drop-jump training which was implemented in a regular school setting. A detrimental effect on jumping performance was observed in the CON group after the first 4 weeks of training, during which augmented feedback was not provided. In contrast, performance in the INT group was not changed after 4 weeks of training. A facilitatory effect from augmented feedback, which has been proven to boost jumping performance in previous studies in adults training 3 to 4 weeks, could thus not be documented in the present study in children. We did not find evidence for positive transfer from drop-jump training to the other motor tasks tested in this study.

### Practical Considerations

The main practical question emerging from these results is: should drop-jump training with augmented feedback be recommended in physical education classes for 4th grade primary school students? Despite showing a positive impact of drop-jump training on drop-jump and countermovement-jump heights, we do not recommend the training for 4th grade physical education classes. The transfer of performance gains to functional motor skills was insignificant. Implementing the training requires resources, and the benefits of the drop-jump training with augmented feedback about the jump height, in our opinion, does not justify the costs of the intervention.

## Figures and Tables

**Figure 1 sports-10-00133-f001:**
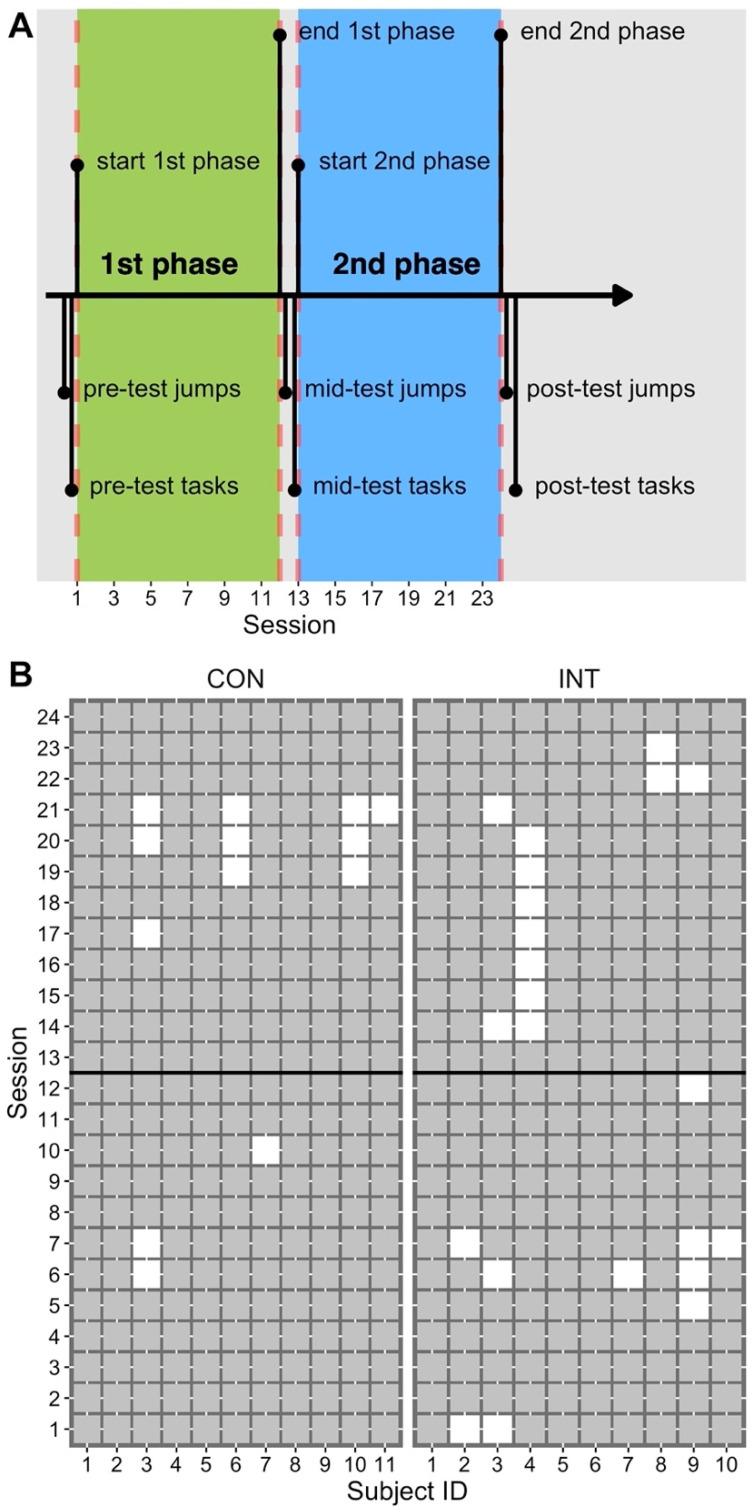
Study design and subject attendance. (Part **A**) depicts the design of the study. First phase (4 weeks, 12 training sessions): children in the INT, but not the CON, group received augmented feedback about the jump height. Second phase (4 weeks, 12 training sessions): all children received augmented feedback about the jump height. Pre-/mid-/post-tests *jumps* refer to the testing of performance of drop-jumps and countermovement-jumps. Pre-/mid-/post-tests *tasks* refers to the testing of performance of rhythmic jumping, standing long-jump, and scissors-jump. (Part **B**): Individual attendance (filled squares) and non-attendance (open squares) of physical education classes/training sessions. Subjects in the CON group missed 4.9% of the session on average, compared to 8.3% in the INT group. There were no dropouts.

**Figure 2 sports-10-00133-f002:**
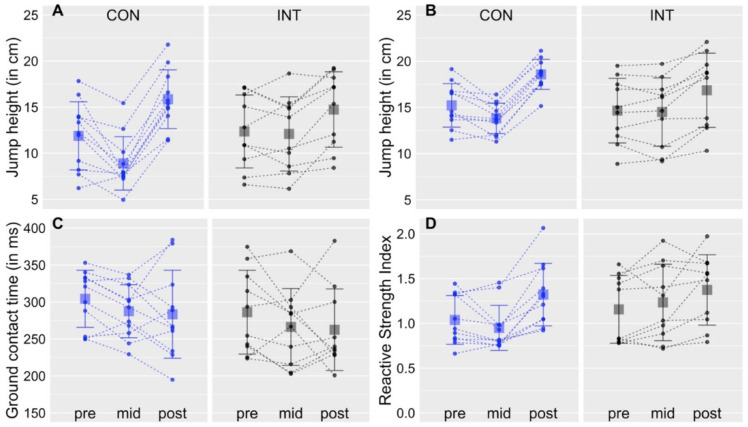
Jumps. Graphs display drop-jump height (**A**), countermovement-jump height (**B**), ground contact times of drop-jumps (**C**), and the reactive strength index of drop-jumps (time airborne divided by the contact time (in seconds)) (**D**), for the pre-, mid- and post-test. Dots represent individual values and squares display group mean values. Vertical lines display the standard deviation of the grand mean.

**Figure 3 sports-10-00133-f003:**
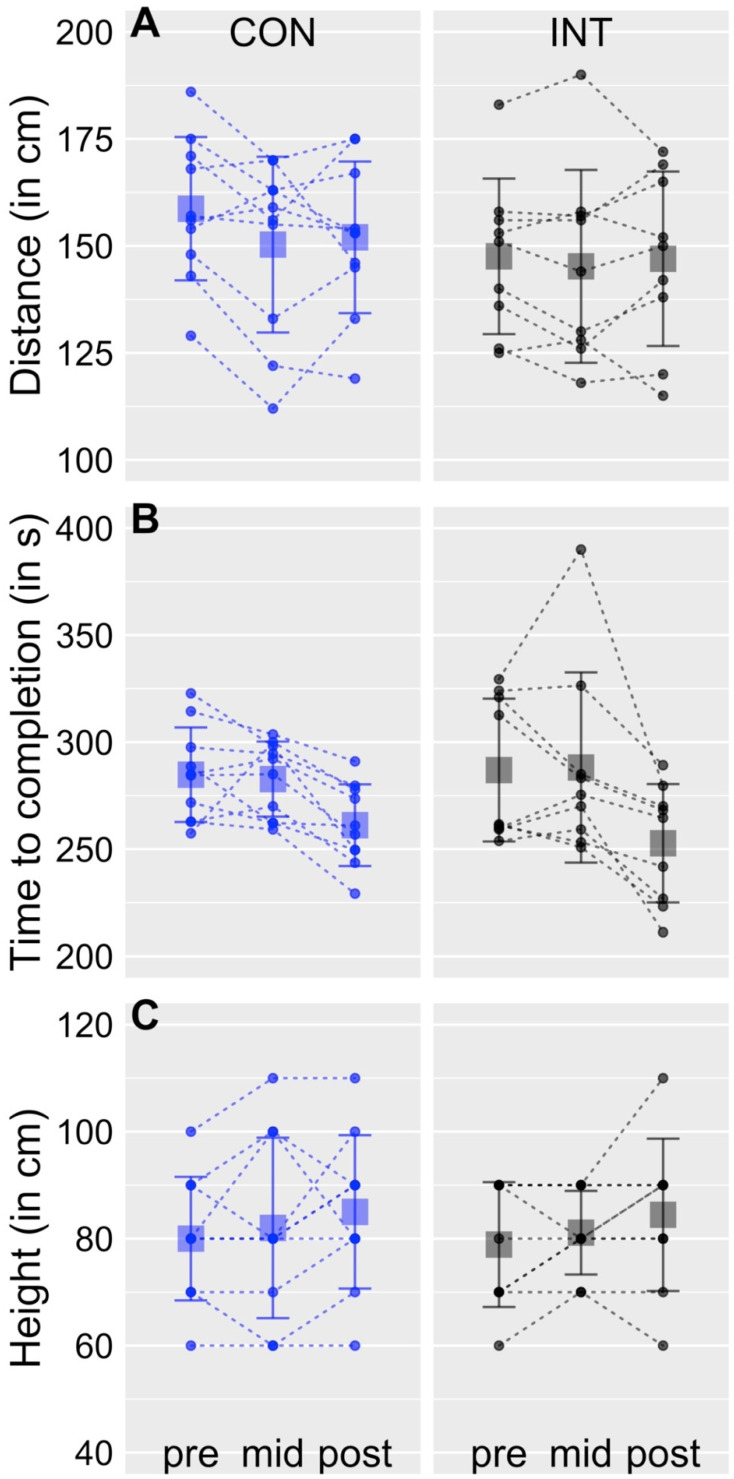
Motor Tasks. Graphs displaying performance of standing long-jump (**A**), rhythmic jumping (**B**), and scissors jump (**C**), for the pre-, mid- and post-test. Dots represent individual values and squares display group mean values. Vertical lines display the standard deviation of the grand mean.

**Table 1 sports-10-00133-t001:** Displays subjects’ characteristics. BMI: The individual weight was divided by the squared individual height. Sex: f—female; m—male.

ID	Age (Years)	Sex	Height (cm)	Weight (kg)	BMI
1	9.7	m	143	34	16.6
2	9.5	m	155	33	13.7
3	9.7	f	149	42	18.9
4	9.2	m	137	29	15.4
5	9.5	m	147	40	18.5
6	9.8	f	139	35	18.1
7	9.6	m	139	34	17.6
8	9.1	m	147	30	13.8
9	9.1	m	152	37	16.0
10	9.3	m	154	34	14.3
11	9.6	m	150	40	17.8
**INT**					
1	9.8	m	148	32	14.6
2	9.8	m	141	30	15.0
3	9.5	f	141	28	14.0
4	9.8	f	151	39	17.1
5	9.4	m	143	30	14.6
6	9.9	m	143	32	15.6
7	9.2	m	149	42	18.9
8	9.1	m	149	34	15.3
9	9.9	m	154	48	20.2
10	9.4	m	145	43	20.4

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
