# Peer review of "Do Primary School Children Benefit from Drop-Jump Training with Different Schedules of Augmented Feedback about the Jump Height?"

_sports, 2022, doi:10.3390/sports10090133_

Round 1
Reviewer 1 Report
Comment: The authors determined the effects of a drop jump program conducted for eight weeks with augmented feedback over the whole program duration (INT group) versus for second half only (CON group) on jump performance in healthy children. Besides others, they found beneficial effects from mid- to post-testing but not from pre- to mid-testing in both groups. The topic of this study is interesting. In addition, the study is well-designed, well-executed, and the results are statistically evaluated in a scientifically solid manner by repeated-measures ANOVA plus post-hoc tests. All results are convincingly explained by the authors, either confirming already existing findings or adding new insights. The conclusions are justified by the results, which are presented numerically and graphically in a comprehensible and adequate way. Since also the manuscript is well structured and written, I found nothing to criticize and recommend the publication of the manuscript in its present form. Maybe the units for height, weight, and BMI could be added to Table 1. Further, the typos “facili-tates” and “ef-fects” (see Limitations section) as well as “det-rimental” should be changed (see Conclusion section).Author Response
Dear Reviewers,
thank you very much for the time you spent on our manuscript. We very much appreciate the comments and think that the quality of the manuscript has been significantly improved based on the suggestions you made. Please find our responses to your comments below. Changes can also be tracked in the attached manuscript.
With the best wishes,
on behalf of all authors
Christian Leukel

Reviewer 2 Report
The present study is of interest to examine the impact of augmented feedback on jumping performance in the 4th grade primary school children executing drop jump training for 8 weeks (24 sessions, 3 16 times/week).
Despite the interesting work, I strongly suggest following the comments to improve the quality of the manuscript. The manuscript needs a written spell check.
Abstract:
1. I suggest adding 95% IC to your results (i.e, for p values and effect sizes).
Material and methods:
2. Please explain how really (i.e, all procedures) students were recruited.
3. L 116. Please, explain why 45 min (3x per week)?
4. L 131. If I understood well, students played in the gym after they were called to be tested in the training science laboratory. If yes, did the authors control what they really played in the gym? If yes, I ask the authors if it is possible that your results were affected by the possible fatigue that was naturally induced during this "played in the gym". Please, clarify.
5. L 158. Regarding the previous commentary (4.). So, you had some children that played more time in the gym, when compared to other children that were firstly tested? Sorry to insist, but I am a little worried that your results could be affected by the possible fatigue that was naturally induced during this "played in the gym".
6. (Figure 1). Authors should indicate % the missing cases and/or dropouts from the initial selection.
7. Statistical procedures might need to be discussed using a within-subjects approach since basic group comparisons were performed.
- How was this comparison attempted? Did the authors pool data for the comparison at the group level? How many data points were paired?
- Can ANOVA for repeated measures test be used (or be appropriate) for such data set?
- Given the high intra-individual variability, a within-subjects approach (recommended for small samples; Linear mixed model [lmm] and generalized linear mixed model [glmm] analysis) might be appropriate (please see some recent works:
https://pubmed.ncbi.nlm.nih.gov/31527865/ , https://pubmed.ncbi.nlm.nih.gov/33672683/; https://www.frontiersin.org/articles/10.3389/fphys.2021.678462/full).
Results
8. Please provide 95 % IC for all effect sizes.
9. Figures 2 and 3: YY axis must start at 0.
Author Response
Dear Reviewers,
thank you very much for the time you spent on our manuscript. We very much appreciate the comments and think that the quality of the manuscript has been significantly improved based on the suggestions you made. Please find our responses to your comments below. Changes can also be tracked in the attached manuscript.
With the best wishes,
on behalf of all authors
Christian Leukel

Reviewer 3 Report
General Comments to the authors:
The main aim of this study was to compare the effect of 8-week (3 sessions per week) using the drop-jump (DJ) exercise using 2 different feedback conditions (“augmented” vs. “control”) on drop-jump and countermovement jump performance. Knowing the chronic effect of manipulating the feedback provided during training sessions in children seems an important aspect, since it has been shown that the presence of the coach in the training sessions and the fact of providing adequate feedback increases the possibilities of improving performance. Therefore, I think that the topic of this paper is appropriate, as there is a need to perform studies in order to address this problematic. However, the rationale of this study, the experimental design and the results are very weak.
As indicated, the current study presents several insurmountable limitations:
1. One of the main limitations is that the sample is too small for a study of these characteristics. In addition, I consider that a control group that does not receive feedback is strictly necessary here to know if the changes in jump performance are due to the feedback provided (either a little or a lot) or to maturational changes typical of age.
2. Reading the methodology section, it seems that everything was done in an improvised way. The weeks of training, the tests used to assess the physical performance, the aims of the study, the procedure used to apply the feedback, etc. The justifications for choosing one test or another or the procedure in which the experimental conditions are applied are very weak and lack scientific rigor.
3. The warm-up protocol performed by the children before performing the jump tests is not described. This is a fundamental aspect to know if the changes are due to the treatment carried out or to other external factors, such as the type of warm-up performed. In addition, no prior familiarization sessions were carried out, which is essential to obtain reliable and valid data.
4. The implemented training program was very poor: 10 DJs for 24 training sessions. It seems a "scarce" training, with little or no progression of the load during the training program and with minimal differences between groups: one group was told the height reached in each attempt and the other was not. What practical utility can be obtained from the results of this study? What is this study for?
5. Finally, although it seems less relevant, Tables and figures are very poor.
Therefore, in my opinion, this manuscript does not fulfill the minimum requirements for publication in this journal.
I am conscious that you have obviously invested a considerable amount of time into this study. However, although I think the topic may be interesting, in the current format and with the methodology used, the document does not meet the minimum quality criteria to be published in a high-impact journal. I wish you all the best and hope that this review does is not received in a too negative light.
Author Response

(The authors gave the same response as above.)

Round 2
Reviewer 2 Report
I am happy with the current version of the manuscript.
The authors did a good job of reviewing the manuscript and answering all the revisions maded.
Reviewer 3 Report
As I mentioned in my previous review, the manuscript contains insurmountable elements that limit its publication. Although the authors have made an attempt to improve the document, there are deficiencies that cannot be solved and that prevent me from giving a positive verdict.